# Primitive Resectable Small Bowel Cancer Clinical–Pathological Analysis: A 10-Year Retrospective Study in a General Surgery Unit

**DOI:** 10.3390/cancers16213713

**Published:** 2024-11-04

**Authors:** Cosmin Vasile Obleagă, Costin Teodor Streba, Cecil Sorin Mirea, Ionică Daniel Vîlcea, Dan Nicolae Florescu, Mihai Călin Ciorbagiu, Tudor Turcu, Mirela Marinela Florescu, Mircea Sebastian Șerbănescu, Alina-Maria Mehedințeanu, Cristin Constantin Vere

**Affiliations:** 1Surgery Department, University of Medicine and Pharmacy, 200349 Craiova, Romania; 2Pneumology Department, University of Medicine and Pharmacy, 200349 Craiova, Romania; costin.streba@umfcv.ro; 3Gastroenterology Department, University of Medicine and Pharmacy, 200349 Craiova, Romania; 4Pathology Department, University Emergency Hospital Bucharest, 050098 Bucharest, Romania; 5Pathology Department, University of Medicine and Pharmacy, 200349 Craiova, Romania; 6Statistics Department, University of Medicine and Pharmacy, 200349 Craiova, Romania; 7Oncology Department, St. Nectarie Oncology Center, 200349 Craiova, Romania

**Keywords:** small bowel cancer, multiple bowel cancer, surgical treatment, immunohistochemistry

## Abstract

Small bowel cancer is considered a rare disease with limited clinical and pathological data but with a rising incidence in recent decades. The imaging, pathological diagnosis, and surgical and oncological treatment, as well as the long-term survival, are variable and related to the pathological type, tumor location, and staging. In our retrospective study, we analyzed a number of patients with primary resectable small bowel cancer who had also presented with exceptional types such as multiple bowel cancers. A total of 46 resectable (R0 resection) small bowel cancer patients were included in this study. Long-term survival depends on tumor aggressivity, invaded lymph node number, and unique or multiple locations.

## 1. Introduction

Small bowel cancer is very rare, but the incidence rate has increased in recent decades (especially carcinoid tumors and adenocarcinoma). The incidence rate is greater in North America and Europe, with a rate of 1.4 in 100.000 inhabitants, which increases after the age of 40. Men have greater incidence rates than women [1]. In 2018, in the United States, the incidence rate was 2.3/100.000 people (0.6% of all cancers diagnosed) [2].

Small bowel cancer risk factors are little known; however, by way of their similarity with colorectal cancer etiology, the following factors can be implicated: inflammatory bowel diseases (Crohn’s disease, ulcerative colitis), celiac disease, small bowel adenomas, familial adenomatous polyposis, Peutz–Jeghers syndrome, alcohol, red meat and smoked goods consumption, as well as Helicobacter pylori and Campylobacter jejuni infection [3]. The low incidence of small bowel cancers can be due to the lower time of exposure of the mucosa to carcinogenic agents because of the faster transit of food through the small bowel [1,4].

Early diagnosis screening tests for small bowel cancer still do not exist, and the diagnosis of uncomplicated duodenal cancer by upper digestive endoscopy is exceptional. Most of these malignancies are diagnosed in complicated forms (bleeding, occlusion, perforation, or local invasion) when surgery is also recommended. In the case of clinical suspicion, the video capsule is indicated for tumors located distally of the duodenojejunal angle.

Surgical treatment of small bowel cancer depends on a series of factors: location, size, and tumor stage, as well as the patient’s biological status. Thus, for the duodenal tumors (duodenum II and III), pancreaticoduodenectomy is recommended, while segmental resections with anastomoses can be attempted for other locations. In concomitant multiple small bowel cancers, multiple resections are recommended if R0 resection can be obtained, and in cancers invading other organs (colon, stomach, etc.), multiorgan resection is recommended if R0 can be obtained. Palliative surgery is recommended in advanced cancers and consists of internal or external digestive derivations [5]. Complete diagnosis of small bowel cancer requires, in most cases, immunohistochemistry associated with conventional histological techniques (and sometimes a second opinion), which guides postoperative systemic treatment [1,4].

The purpose of our study was to analyze the clinical and pathological characteristics of a primitive small bowel cancer patient lot, managed in a surgical unit in a tertiary care hospital over 10 years (February 2014–February 2024).

## 2. Materials and Methods

We performed a retrospective study including 46 patients diagnosed with duodenum, jejunum, and ileum complicated and uncomplicated cancers, hospitalized in the II^nd^ Department of General Surgery, Clinical County Emergency Hospital of Craiova, between February 2014 and February 2024. The following data were analyzed: demographic data, clinical and imaging aspects (CT, MRI), intraoperative aspects, surgical technique, and conventional anatomopathological and immunohistochemical results. The inclusion criteria were patients over the age of 18 diagnosed with primitive small bowel cancer, including multiple location cancer confirmed through histopathological exam (which provided information about the tumor type, differentiation degree, vascular or nerve invasion, and pTNM classification); the exclusion criteria were the presence of distant metastasis and advanced inoperable bowel tumors. The diagnosis was confirmed in all cases through histopathological and immunohistochemical evaluation (except for patients who died immediately after surgery where immunohistochemistry was not performed because of lack of consent from the family), and, in approximately 25% of cases, a second anatomopathological opinion was needed.

The Clinical County Emergency Hospital of Craiova Ethics Committee was informed, and this study was approved (no. 17886/24 April 2024) on the following bases: (1) data were collected within a retrospective, observational study; (2) the study did not interfere with current medical care; (3) patients were not treated with experimental substances, and no biological samples were taken during the study; and (4) data were collected and analyzed anonymously so that the patient data confidentiality would not be breached. The following variables were collected: (1) patient general data (age, sex); (2) diagnosis method (through imaging exams or intraoperatively); (3) tumor characteristics (tumor location, complication type, surgery type); (4) resected tumor histopathological characteristics and tumor stage; (5) long-term survival.

The tumor stage was assessed following the eighth edition of the “TNM Malign Tumor Classification” from the International Cancer Control Union [6]. The immunohistochemical panel applied in all cases (except for those with no informed consent) excludes malignant melanoma (Melan A, HIMB45), solitary fibroid tumors with an extensive adipocyte component (STAT 6 immunophenotype), metastases that express synaptophysin or extrarenal angiomyolipoma (enterally) through SMA immunophenotyping, focal Desmin and h-Caldesmon, and also the evaluation of endothelial differentiation (CD31, CD34) or tumor proliferation index ki67. Final TNM staging, grading, and vascular and perineural invasion were also analyzed, together with the IHC (immunohistochemical) profile, which included, depending on the case, ANTI-CDX2, KI-67, CD56, ANTI-p53 EA, ANTI-CD10, DOG-1, and S100 ANTIBODY (all immunohistochemical reagents were provided by the Thermo Fisher Scientific LSG, Waltham, MA, USA). Immunohistochemistry was also needed to diagnose small bowel melanoma or occult cancer bowel metastases (the cases being excluded from the study). Patients received standard oncological treatment according to guidelines, or treatment personalized by the oncological committee, and remote tracking was conducted by the oncologists. We used Microsoft Excel 2019 MSO (version 2304 Build 16.0.16327.20200) to build a comprehensive database, in which we included all the variables, and MedCalc version 20.218 software was used for statistical analysis. The frequencies were presented as absolute numbers of cases and percentages. Chi-squared tests were used to compare ordinal or nominal variables. Continuous variables were compared using the U Mann–Whitney test if the variable was not normally distributed. A *p*-value smaller than 0.05 was considered statistically significant.

## 3. Results

A total of 64 small bowel cancer patients were included in the cohort of this study. Of these, 18 patients were excluded according to the exclusion criteria (distant metastasis, advanced unresectable intra-abdominal disease), and 46 patients were included in the final analysis (Table 1).

Regarding gender distribution, in our study, we found a male predisposition for small bowel cancer (30 men vs. 16 women), and the most affected age group was older than 70 years old (23 cases). The average age of the patients included in this study was 66.4 ± 11.7 years. Even though the patient group was relatively small, over the 10 years, a progressive increase in small bowel cancer incidence was noted, especially in recent years, except for the year 2020, which corresponded to the onset of the COVID-19 pandemic. 

In most patients included in this study (43 patients—93.4%), the small bowel tumors were diagnosed during an acute or progressive complication, such as occlusion through stenosis (25 patients—54%) or intussusception (3 patients—6.5%), bleeding (10 patients—21%), perforation (3 patients—6.5%), or local invasion. Local invasion in other organs (two patients) manifested clinically as a chronic occlusion or as a false diarrhea syndrome secondary to an entero-colic fistula and was explored through paraclinical imaging exams; in two other patients, diagnosis was established through a CT exam recommended for nonspecific symptomatology. Regarding tumor location, in our study, the main location was in the ileum (23 patients—50%), followed by the jejunum (32.60%), duodenum (8.6%), and the duodenojejunal flexure (Figure 1). One of the most interesting findings in our study was the presence of multiple tumors in the small bowel (four cases with multiple locations along the small bowel, and two to six tumors in each case). Although these tumors cannot be considered synchronous (they contain the same histological type), their position as small bowel primitive tumors or metastases is questionable, and no consensus has yet been reached (Figure 2a,b).

Regarding small bowel cancer diagnosis in the complication phase, it was mainly intraoperative in the first 3 years, but because of an increased CT scan usage in the emergency room in recent years (since 2020), small bowel cancer diagnosis was more often established preoperatively (Figure 3). The CT aspect in small bowel cancers can vary from bowel occlusion because of a stenosing tumor or intussusception, to bleeding or uncomplicated tumors (more frequent in the case of GIST). A CT exam was recommended in the emergency department for 16 patients, for 15 (93.3%) of whom the diagnosis of small bowel tumor was established (Figure 4b–d). Esophagogastroduodenoscopy (EGD, Figure 4a) was the preferred method for the diagnosis of duodenal tumors (four patients), and tumors located at the duodenojejunal angle (four patients), both as a primary method and also as a complementary method in the case of clinical or imagistic suspicion. Upper digestive endoscopy permits the visualization and biopsy of the lesion and also has a therapeutic role through tumoral stenting (duodenal tumoral stenting was performed in three patients with stenosis before the surgical step). The association of an endoscopic ultrasound may be useful in duodenal cancers for assessing in-depth invasion (pancreas, superior mesenteric pedicle). Although useful, the video capsule did not diagnosed any small bowel tumors in our patients. When acute complications (bleeding, occlusion, or perforation) were absent, MRI was recommended (Figure 5a,b) for the assessment of the possibility of resection (four cases of locally invading tumors). Intraoperative aspects of several small bowel tumors are presented in Figure 6, Figure 7 and Figure 8.

From 46 small bowel tumors, 26 (56.52%) cases were represented by adenocarcinomas, 8 (17.39%) cases were represented by GISTs, 11 (23.9%) cases were represented by lymphomas, and a single case (2.17%) was represented by a neuroendocrine carcinoma. Out of the total 27 bowel adenocarcinoma cases, 5 cases were categorized as well differentiated, 10 cases were categorized as moderately differentiated, and just 12 were represented by poorly differentiated tumors. Most of the cases were diagnosed in advanced stages of the disease: 15 cases were included in stage III (9 cases IIIA and 6 cases IIIB, respectively) and 8 cases in stage IIA; in comparison, only 4 cases were included in stage I. Vascular invasion was present in 21 adenocarcinoma cases and perineural invasion in 16 cases. Also, out of the 27 adenocarcinoma cases in total, 26 were conventional-type NOS adenocarcinomas and just a single case was represented by a sarcomatous carcinoma. From the immunohistochemistry point of view, all cases expressed CK (AE1/3). The immune marking for p53 was intensely positive in 19 cases and negative in 8 cases; the Ki67 marked cells were between 10 and 55%, with an average of 27%.

Regarding the pT category, most adenocarcinomas were advanced T3 (15 cases), T4 (4 cases), and T2 (7 cases), and tumors in low stages were also found. Concomitant with extension through the colon wall layers, tumor aggressivity increased, as did vascular and perineural invasion.

The average number of recovered lymph nodes was greater than eight, but it is worth mentioning that duodenojejunal flexure adenocarcinomas (four cases) raised the biggest problems concerning the possibility of obtaining this number and, in the case of terminal ileum carcinomas (when a right ileocolectomy was recommended—three cases), the number of recovered nodes was significantly higher (22 in comparison to 8). 

In the case of gastrointestinal stromal tumors (eight total cases analyzed), six cases were included in low-degree tumors (G1) (less than five mitoses/10HPF), and just two cases were included in high-degree tumors (G2) (more than five mitoses/10 HPF). Vascular invasion was present in four cases, and perineural invasion was present in just two cases. From an immunohistochemistry point of view, all cases were positive for DOG1, CD34, and CD117 and negative for other mesenchymal markers (s100, SMA). The Ki67 immunostaining was positive in an average of 11% of cases. The malignancy risk was intermediate.

Regarding the histopathological result, duodenal-located cancers were diagnosed preoperatively through a histopathological exam of the tumor tissue obtained through EGD. All jejunum and ileum cancers were diagnosed postoperatively from the resection piece and locoregional lymph nodes, and in all cases, immunohistochemistry was needed to diagnose or confirm. In the studied patient group, we encountered a very rare case of multiple location small bowel sarcomatous carcinoma, and the immunohistochemistry was definitive in formulating the diagnosis (Figure 9, Figure 10, Figure 11 and Figure 12) by excluding a malign melanoma, an epithelioid gastrointestinal tumor, neuroendocrine tumor, or a metastasis; thus, the deficit of SMARCB1/INI1 of an undifferentiated carcinoma supported the sarcomatous carcinoma diagnosis.

From a statistical point of view regarding the histological report, it was found that the most frequent complication of adenocarcinomas was stenosis; of lymphoma, it was bleeding (*p* = 0.01576); and of sarcomas, it was both bleeding and stenosis (Figure 13 and Figure 14).

Regarding long-term survival, we have found that long-term evolution is related to the aggressivity of the tumor (G1/G3), to the presence of more tumors, and invariably to the type of tumor resection. Considering that the long-term evolution depends on resectability and its type, and that, in our study, we did not include incomplete resection patients (R1 or R2), we took into account the presence of more tumors, tumor aggressivity (G), and lymph node status (N). The median period of tracking of the entire lot was 25 months, and Figure 15 presents the Kaplan–Meyer curve related to the tumor uniqueness, aggressivity, and the invaded lymph node number. Seven patients (15.21%), of which five had adenocarcinomas, one GIST, and one lymphoma, surpassed 5 years of monitoring without observing disease evolution, all had a moderate differentiation degree (G2), and the tumors were unique and without locoregional metastases (N0). Twelve patients (26.1%) surpassed 4 years of monitoring. During monitoring, 15 (32%) patients died, but only in 11 (23%) patients was a tumor-related cause of death identified. Because of the relatively low number of patients, statistical significance was not achieved, but survival was clearly bound to the number of tumors (unique/multiple), tumor differentiation degree, and number of invaded lymph nodes.

Because, in our study, the most frequent small bowel tumor was represented by adenocarcinoma, we also analyzed the Kaplan–Meyer curve for adenocarcinoma. Similar findings regarding the correlation with tumor uniqueness, grading, and invaded lymph nodes were noted, with similar limits regarding the small number of patients.

## 4. Discussion

Small bowel cancer is a rare pathology with a late diagnosis, most of the time in acute complication stages, and, in comparison with colorectal cancer, there still are no screening tests for early diagnosis; the EGD diagnosis of uncomplicated duodenal cancer is exceptional. Most of the malignancies are diagnosed in complicated forms (bleeding, occlusion, perforation, or local invasion) when surgery is also recommended. In case of clinical suspicion, the video capsule is useful for tumors located distally of the duodenojejunal angle [7,8,9].

Similarly to the literature data, our study reveals the rarity of this pathology; most of the studies contain a limited number of patients, and publications with large patient groups are exceptional [10,11]. An increased incidence is noted in males over 60 years old, with the ratio between males and females varying between 1.5 and 2 vs. 1; small bowel cancer can rarely be diagnosed in young adults (under 50 years). In the current study, the gender ratio is approximately 2:1 in favor of males, and general incidence tends to rise over time; similarly, the data from the USA and Europe have shown that the number of cases has doubled in the last 40 years in developed countries [2].

The diagnosis of small bowel pathology changed because of the evolution of invasive and noninvasive imaging methods; in the past, the intraoperative diagnosis was a rule, but currently, the CT scan and enteroscopy seem to be the most used methods of diagnosis. It is considered that the rate of detection for bowel tumors through a CT scan is 70–80% of all small bowel cancers but can be even greater depending on the tumor location and the evolutionary character (in the complication phase) [12,13]; stenosing small bowel tumors manifested with occlusion have a higher diagnostic rate than uncomplicated tumors. By comparison, a study published by Yoo AY and collaborators in 2021 analyzed 510 patients with miscellaneous abdominal symptoms (abdominal pain, occult bleeding), which were assessed with the help of videocapsule and balloon-assisted enteroscopy over 9 years (January 2010–September 2018); the diagnosis of small bowel cancer was established in 28 patients [8]. In our study, the diagnosis was established in most cases intraoperatively (56.52%) in patients with an acute abdomen clinical and paraclinical picture (occlusion, digestive bleeding, perforation); stenosis was the most common complication for adenocarcinoma, while bleeding was mostly seen in lymphomas and sarcomas. CT scan has diagnosed 32.6% of the small bowel cancers, surpassing the intraoperative diagnosis in recent years (Figure 3). Similarly to our study, duodenal and duodenojejunal flexure cancers were most frequently diagnosed through EGD, recommended in patients with clinical symptomatology or imagistic suspicion [14]. Although there is the possibility of assessing the small bowel with the help of the video capsule, in our study, we did not include any patient diagnosed through this method because the patients were hospitalized in a surgery unit because of a previous bowel tumor diagnosis or a high suspicion thereof with laparoscopy/laparotomy recommendation. Generally, MRI imaging is recommended in cases of bowel cancer suspicion before enteroscopy. By providing refined information and MR, enteroclysis is considered the best examination method for the small bowel [15]. In our study, this method was used in assessing tumors with locoregional invasion (four patients) to establish the resectability after an initial CT scan, but not in case of an acute progressive complication (occlusion, bleeding, perforation).

Intraoperative diagnosis was made in most patients with complicated small bowel cancer with a jejunum or ileum location, and R0 resection (if it is possible) is the rule. The intraoperative diagnosis of the tumor lesion requires an extemporaneously pathological exam recommended for the indication of lymphadenectomy or, in multiple locations, for the indication of resection of more bowel segments. For duodenal tumors, the natural evolution is towards occlusion and less towards hemorrhage. Out of the four patients with duodenal cancer, three of them presented signs of a small bowel occlusion requiring the placement of a trans-tumoral stent through EGD (two cases), and in one case, a tumoral by-pass was performed (gastro-enteric anastomosis), having the goal of increased performance status (curative surgery) and, respectively, pancreaticoduodenectomy. Locoregional advanced small bowel cancers benefited from multiorgan resection (Figure 7) when R0 resection was possible. Despite being intraoperative in all cases, they were considered different-location tumors with bowel invasion, and the anatomopathological result reconsidered the primary location at an enteral level. The enteral resection level was imposed by the tumor limit and remaining bowel vascularization. Choosing the section limits (difficult in the case of the proximal jejunum) took into account the obtaining of a negative oncological resection edge and bowel vascularization. Nevertheless, even though for a correct disease staging (TNM classification) the assessment of at least eight locoregional nodes is necessary [7], in some cases, this objective was difficult to achieve in our study.

Surgery is the most important step in managing small bowel cancer, having the role of confirming the preoperative diagnosis, and it is the only one that can initiate the cure, with the main goal being R0 resection, which is possible in approximately 65% of the cases [16,17]. Generally, for duodenal cancers, surgery is performed electively after hydro-electrolytic rebalancing and detailed general exploration, and in the case of ileum and jejunum cancers, emergency surgery is most frequently performed. In the case of duodenal cancer, the most recommended surgery is represented by the pancreaticoduodenectomy, specifying that in the duodenum I or IV location, an R0 edge resection can be performed. Similarly to the literature data [18,19,20], our report offers similar information regarding the duodenal location (four cases), as well as the recommendation of curative pancreaticoduodenectomy in three cases. For duodenojejunal flexure cancers, surgery can encounter difficulties in obtaining the proximal R0 resection limit and restoring intestinal transit [21]. The literature data are limited and refer to a series of cases or case reports, highlighting the rarity of the pathology and the surgical technical difficulties regarding the resection and restoration of the intestinal transit [21,22]. The impossibility of obtaining regional lymph nodes (eight minimum) en bloc with the operative piece and the rapid metastasis in the superior mesenteric nodes are other problems in the correct diagnosis (pTNM) and evolution of cancers in this location. In our study, four cases were included (two adenocarcinomas, one GIST, and one lymphoma), of which three cases were resectable, and in the case of one adenocarcinoma, an internal derivation was performed because of the patient’s altered general state.

Generally, jejunum and ileum cancers (up to approximately 10 cm from the ileocecal valve), if they are resectable, do not pose important problems, but their multiple locations along the small bowel require special attention. Because a complete classification of these tumors that do not respect the synchronous term still does not exist, they cannot be considered metastases, but from the reduced report from the literature, the etiopathogeny seems to have a genetic determinism [23]. Sometimes, the heterogeneous clinical symptomatology and unclear imagistic aspect can lead to a wrong interpretation and a late diagnosis [24,25]. Also, malignancy needs to be confirmed through an extemporaneous pathological exam for curative surgery, and the invalidation of the malignancy contraindicates lymphadenectomy and thus avoids an extended resection. 

In our study, we have included four cases of multiple tumors, all stenosing adenocarcinomas (Figure 2), in which multiple segment enterectomies were performed, even though R0 resection was very difficult to perform because of the presence of mesenteric blocks of lymph nodes (sometimes they impose the extirpation of a greater length of the bowel). Especially in small bowel cancers with multiple locations, the enterectomies can be multiple, or a bowel segment that contains more tumors can be removed, with special attention to the remaining bowel length (short bowel syndrome). Long-term survival in the four cases was between 4 and 16 months, even though resection was considered R0, and the cytology exam was negative. Histopathology exam revealed tumor aggressivity through N>2 and G>3 in these patients, with survival under 11 months. In evolution, distant metastases and tumor aggressivity negatively influenced survival. In tumoral locations close to the ileocecal valve, the right hemicolectomy is recommended [26], with the ileocecal appendicular trunk resection in its origin in the mesenteric vessels and lymphadenectomy at this level. In our study, all terminal ileum tumors manifested clinically as a bowel occlusion, and all benefited from a right hemicolectomy without postoperative complications.

A total of 43 patients (93.4%) were diagnosed in the complications phase, and the histopathology exam was in concordance with tumoral staging; therefore, 15 cases were stage III (9 cases IIIA and 6 cases IIIB), 8 cases were stage IIA, and only 4 cases were stage I, similar to other studies [27].

Few of the studies reported showed a connection between the histological type and the complications of the tumor [8,11,28]. In our study, we found a statistically significant relation (*p* = 0.01576) regarding the pathological type and the presence of complications, and also that the most frequent complication was represented by stenosis for adenocarcinomas, bleeding for lymphomas, and both bleeding and stenosis for sarcomas and GIST. The conventional anatomopathological exam is frequently insufficient for a complete diagnosis of small bowel cancer, and the immunohistochemistry exam is recommended, similar to other rare cancers [29,30]. Complex operative pieces (multiorgan) require supplementary markings for a more complete diagnosis. In our study, the immunohistochemistry exam was performed for all patients (except for those who died early preoperatively), and because of the rare forms in 11 of the patients (25%), a second anatomopathological opinion was needed to reconfirm the diagnosis.

Regardless of resection type (R0/R1 or R2), several studies [27,31,32,33] have shown that advanced age, tumor markers (CEA, CA19-9), duodenal location, higher grading, pT4 stage, positive lymphovascular invasion, and the number of positive lymph nodes are negative prognostic factors in small bowel cancer patients. In our study, grading, the number of invaded lymph nodes, and the number of tumors (greater than 1) were independent prognostic factors for survival. The duodenal location of the tumor, when pancreaticoduodenectomy was performed (four patients), did not influence survival, with all patients surviving (monitoring period between 1 and 5 years). Some studies [31] did not differentiate between pure duodenal location and duodenojejunal flexure level location [21]; however, in our study, out of the four patients with this location, three died in an interval between 1 and 5 years, suggesting a higher mortality rate in those cases.

Several published studies from Romania have analyzed small bowel tumors. Two videocapsule-based studies in an academic center found a total of 19 confirmed small bowel tumors from 404 examinations (detection rate 4.7%) for 15 years [34,35], and another small videocapsule-based study of 29 examinations over one year found 10 small bowel tumors (34.5%) [9]. Several surgery-based studies are available: a study included 73 small bowel tumors operated over 12 years, 71.2% being malignant, and a proportion of 11.5% of small bowel tumors from the operated digestive tumors [36]; another study found 57 small bowel malignant tumors over 15 years, with GIST being the most frequent (42.1%), followed by adenocarcinoma (33.3%), lymphoma (14%), and neuroendocrine tumors (3.5%) [26]. Another surgical study over 12 years found 31 small bowel tumors, of which 25 were malignant; adenocarcinoma was the most frequent malignant tumor (32%), followed by GIST (28%) and lymphoma [37]. In an 18-year study, 63 patients with small bowel tumors were operated on, from which 45 (71.4%) were malignant; of these, 53.3% were carcinomas, 22.2% were lymphomas, 10% were stromal tumors, and three cases were metastatic melanomas [38]. Although the most frequent small bowel malignancies were adenocarcinoma, stromal tumors, and lymphoma, the proportion slightly differed between these surgical series. 

The limitations of this study derive from the fact that it is retrospective, from a single department, and has included a relatively small number of bowel cancer cases because of the rarity of this disease. By including data from more centers, a higher number of analyzed cases may be possible and, as a consequence, the ability to formulate more statistically significant conclusions may increase. An accurate assessment of survival, however, remains a difficult goal because a long period of surveillance may be needed. 

## 5. Conclusions

The results of this study reveal that small bowel cancer poses significant diagnostic, imaging, and surgical difficulties, particularly in cases with simultaneously multiple locations of small bowel cancers, an exceptional form associated with a rare pathology. Negative prognostic factors, independent of the resection type, are tied to the tumor aggressivity, the invaded lymph node number, and the unique or multiple tumor locations.

## Figures and Tables

**Figure 1 cancers-16-03713-f001:**
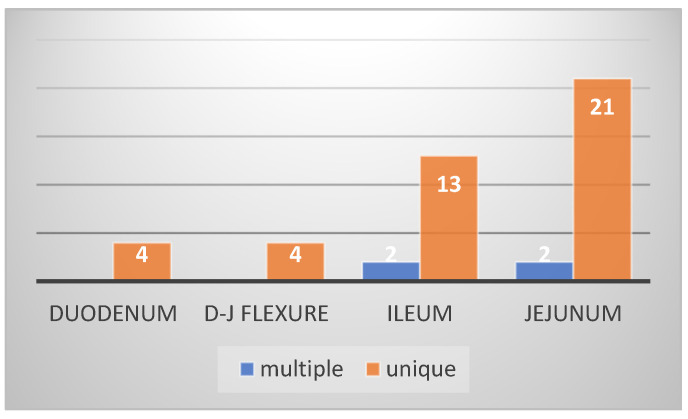
Topographic distribution of tumors.

**Figure 2 cancers-16-03713-f002:**
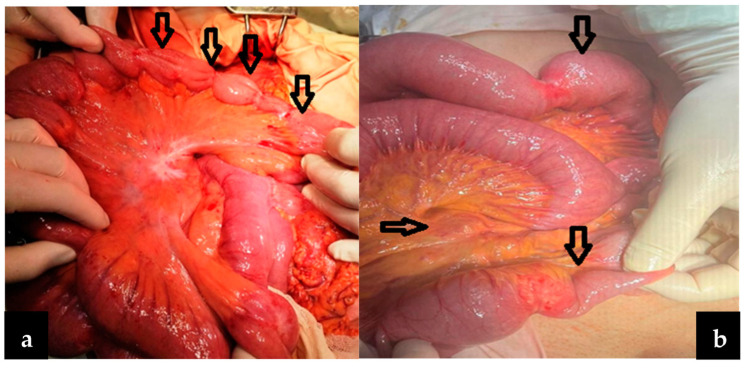
(**a**,**b**) Intraoperative aspect: multiple location bowel carcinoma (arrows).

**Figure 3 cancers-16-03713-f003:**
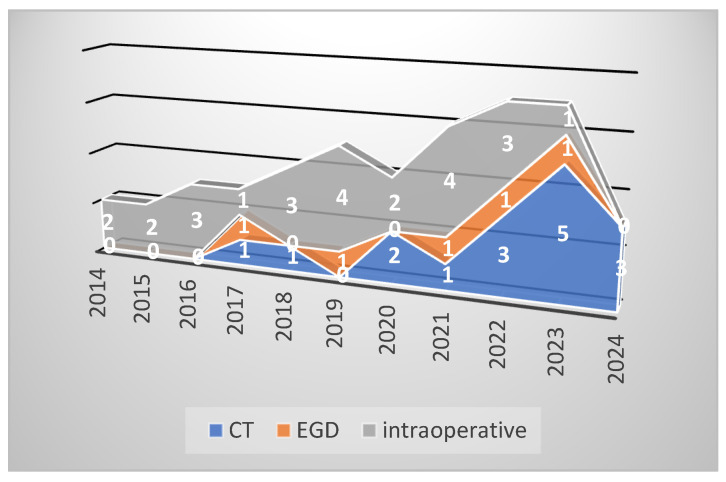
Small bowel cancer diagnosis: intraoperative vs. CT vs. EGD.

**Figure 4 cancers-16-03713-f004:**
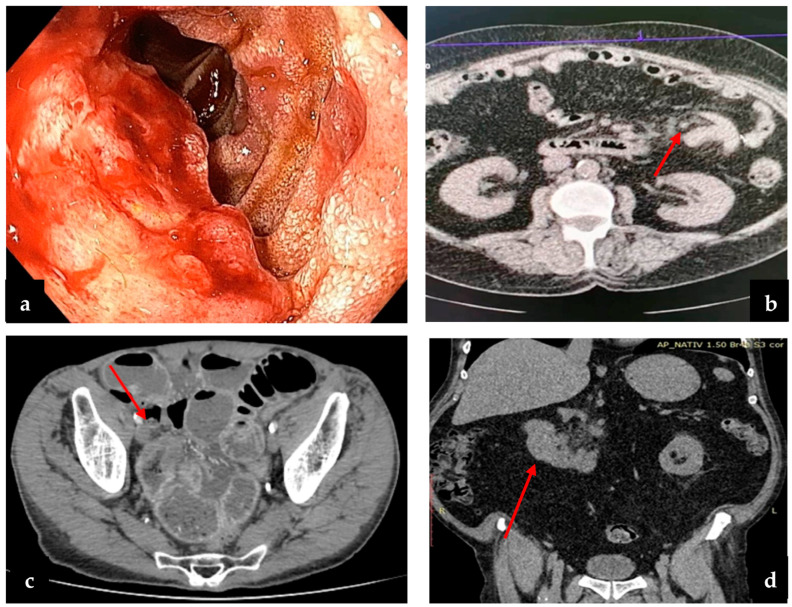
(**a**) Endoscopic examination of a duodenal tumor; (**b**) CT exam of tumor bowel intussusception; (**c**) postcontrast CT scan, arterial phase, axial plane—there is a marked stenosing circumferential parietal thickening of a pelvic small bowel loop (arrow); (**d**) CT exam of duodenal tumor and perilesional lymph nodes.

**Figure 5 cancers-16-03713-f005:**
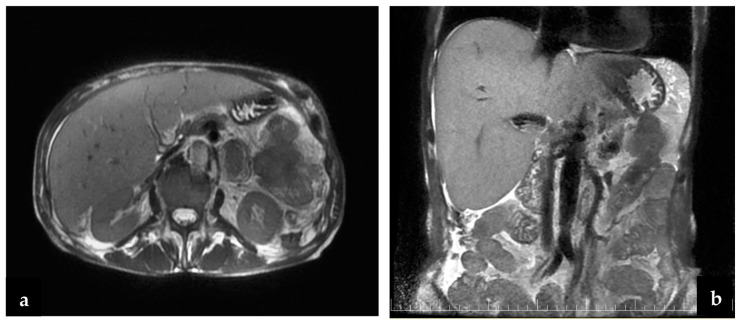
(**a**,**b**) Abdominal MRI: the tumor includes the first jejunum loop and the transverse colon and invades the gastric wall.

**Figure 6 cancers-16-03713-f006:**
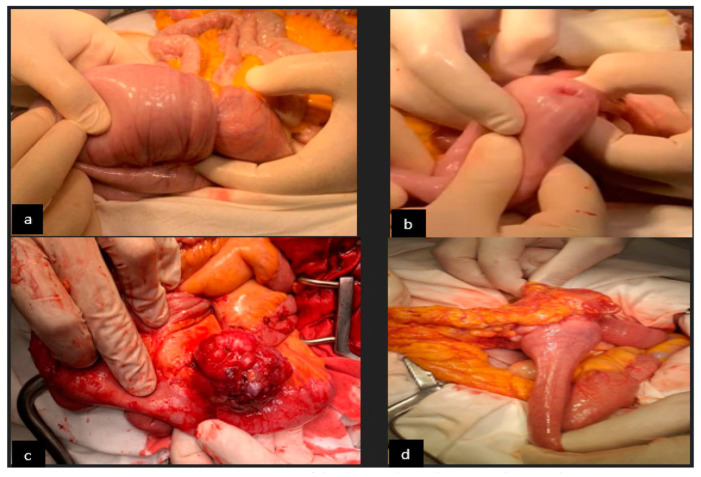
Intraoperative aspect: (**a**,**b**) bowel occlusion through jejunum tumor intussusception; (**c**) ileum GIST; and (**d**) small bowel infiltrative carcinoma with great omentum invasion.

**Figure 7 cancers-16-03713-f007:**
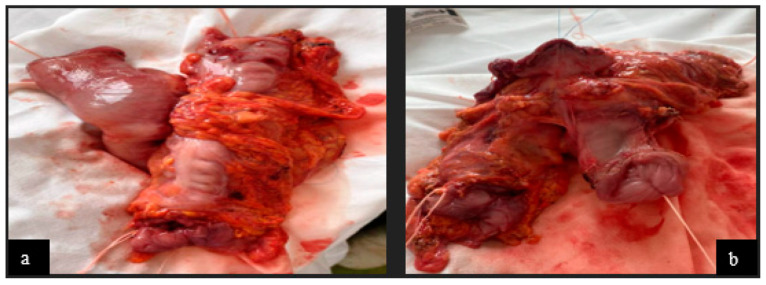
Operating fragment: (**a**) jejunum adenocarcinoma with invasion in the transverse colon and posterior gastric wall—front view and (**b**) posterior view.

**Figure 8 cancers-16-03713-f008:**
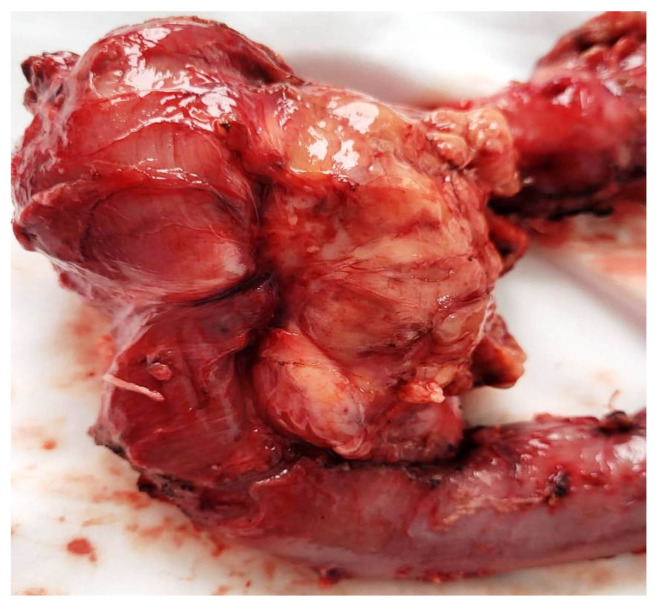
Pancreaticoduodenectomy piece: duodenum II cancer with invasion in the head of the pancreas.

**Figure 9 cancers-16-03713-f009:**
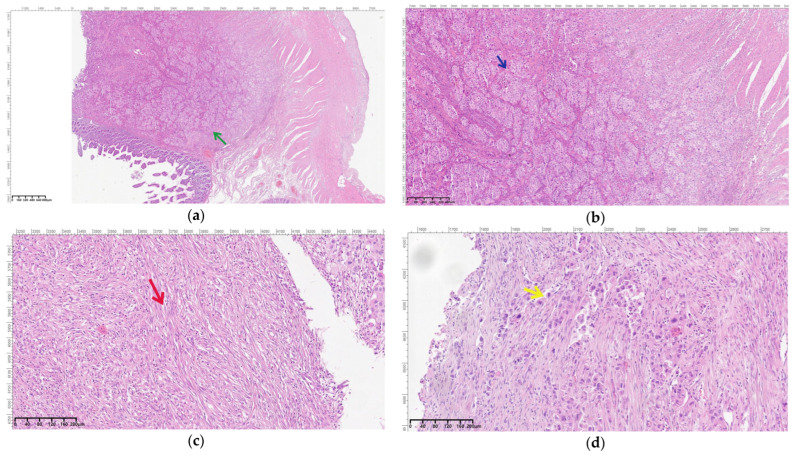
Malignant proliferation with mostly solid pattern infiltrating the small intestine wall ((**a**)—green arrow; original magnification 40×) composed of epithelioid cells arranged either solitarily or in nests and cords ((**b**)—blue arrow; original magnification 100×) and spindle cells ((**c**)—red arrow; original magnification 200×). The cells are markedly pleomorphic, with anisonucleosis, frequent atypical mitoses, and areas of tumor necrosis ((**d**)—yellow arrow; original magnification 200×).

**Figure 10 cancers-16-03713-f010:**
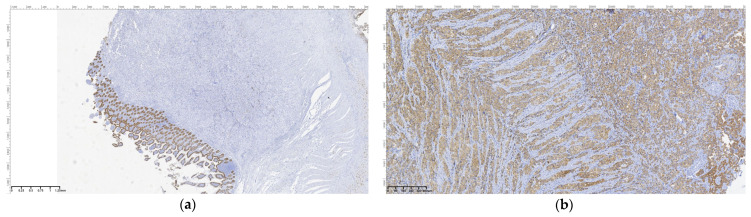
Sarcomatous carcinoma immunohistochemical diagnosis: (**a**)—Oscar—OM 40×; (**b**)—cytokeratin 8/18 positive in epithelioid cells, OM 100×; (**c**)—cytokeratin 8/18 positive in spindle cells, OM 200×; (**d**)—S100, OM 200×.

**Figure 11 cancers-16-03713-f011:**
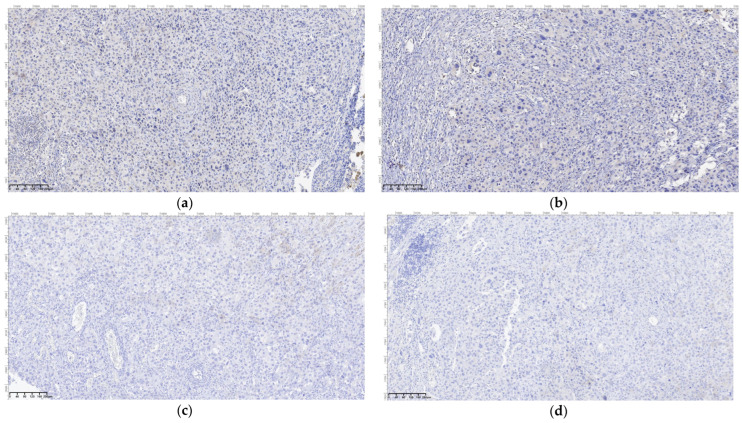
Sarcomatous carcinoma immunohistochemical diagnosis: (**a**) SOX 10, OM 200×; (**b**) PRAME, OM 200×; (**c**) MelanA, OM 200×; (**d**) HMB45, OM 200×.

**Figure 12 cancers-16-03713-f012:**
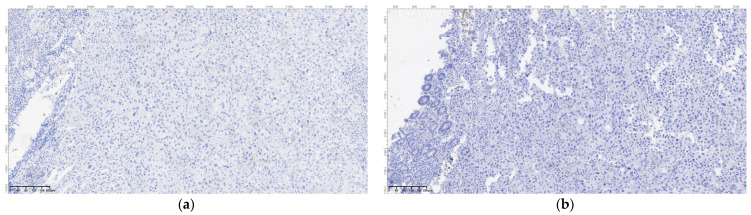
Sarcomatous carcinoma immunohistochemical diagnosis: (**a**) DOG1, OM 200×; (**b**) synaptophysin, OM 200×; (**c**) SMARCB1; OM 200×; (**d**) BRAF V600E IHC; OM 200×.

**Figure 13 cancers-16-03713-f013:**
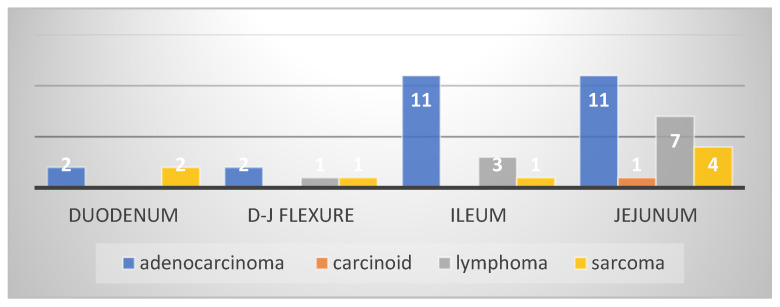
Small bowel cancer distribution related to the topography and pathology.

**Figure 14 cancers-16-03713-f014:**
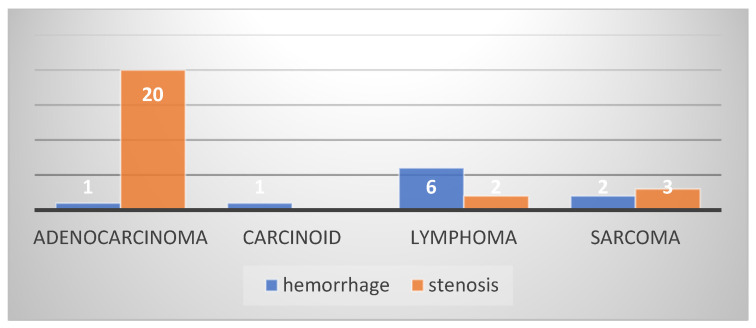
Complications in small bowel tumors related to the pathological type.

**Figure 15 cancers-16-03713-f015:**
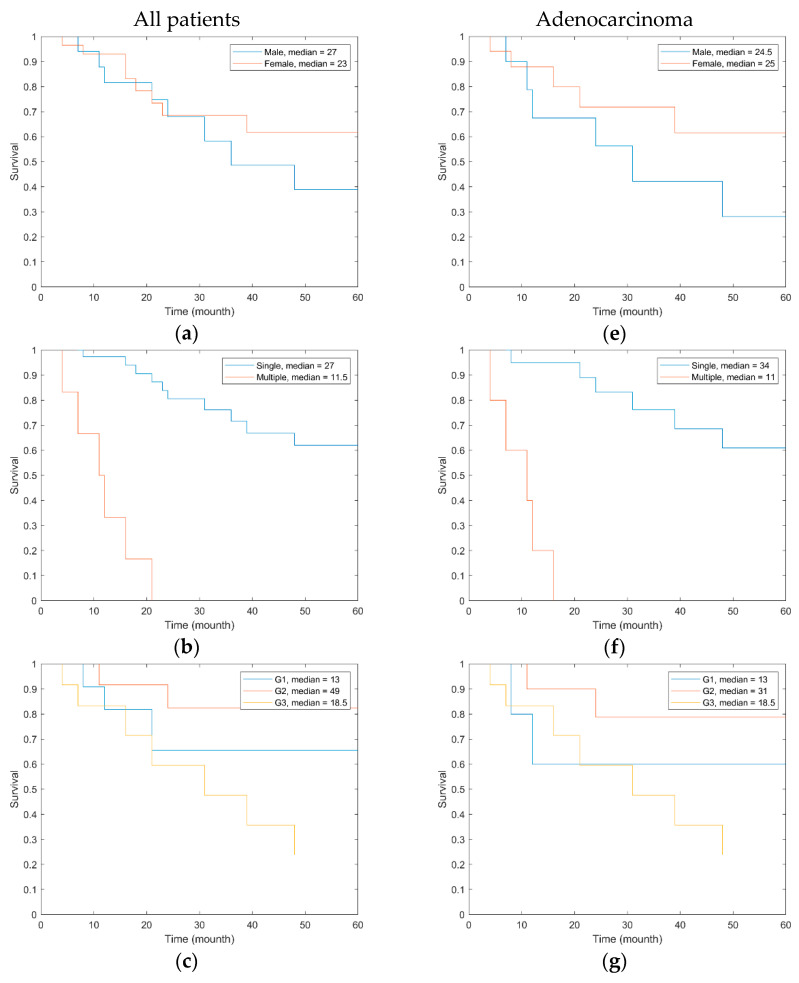
The survival rate for all small bowel tumors depends on (**b**) tumor number—unique vs. multiple; (**c**) tumor grading; (**d**) number of invaded lymph nodes; (**a**) no relation with gender being found. For small bowel adenocarcinomas, the survival rate depends on (**f**) tumor number—unique vs. multiple; (**g**) tumor grading; (**h**) number of invaded lymph nodes; (**e**) no relation with gender being found.

**Table 1 cancers-16-03713-t001:** The patient groups included in this study.

Gender	
Men/women (%)	30/16 (65.22/34/78)
Year of the diagnosis	
- 2014	2 (4.35%)
- 2015	2 (4.35%)
- 2016	3 (6.52%)
- 2017	3 (6.52%)
- 2018	4 (8.7%)
- 2019	5 (10.87%)
- 2020	4 (8.7%)
- 2021	6 (13.04%)
- 2022	7 (15.22%)
- 2023	7(15.22%)
- 2024	3(6.52%)
Diagnosis	
- CT scan	15 (32.60%)
- upper digestive endoscopy	5 (10.86%)
- intraoperatively	26 (56.52%)
Tumor location	
- duodenum	4 (8.69%)
- duodenojejunal flexure	4 (8.69%)
- jejunum	15 (32.60%)
- ileum	23 (50%)
Number of tumors	
Multiple	4 (8.7%)
Unique	42 (91.3%)
Surgery	
- by-pass/resection	2 (4.34%)
- pancreaticoduodenectomy	3 (6.52%)
- enterectomy	33 (71.73%)
- multiple enterectomies	3 (6.52%)
- right ileocolectomy	3 (6.52%)
- multiorgan resection	2 (4.34%)
Tumor type	
- adenocarcinoma	26 (56.52%)
- carcinoid	1 (2.17%)
- lymphoma	11 (23.9%)
- sarcoma	8 (17.39%)
Complications at the time of diagnosis	
- no complication	3 (6.52%)
- bleeding	10 (21.73%)
- intussusception	3 (6.52%)
- local invasion	2 (4.34%)
- perforation	3 (6.52%)
- stenosis	25 (54.34%)

## Data Availability

The data presented in this study are available from the corresponding author on reasonable request.

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
