# Peer review of "Primitive Resectable Small Bowel Cancer Clinical–Pathological Analysis: A 10-Year Retrospective Study in a General Surgery Unit"

_cancers, 2024, doi:10.3390/cancers16213713_

Round 1
Reviewer 1 Report
Comments and Suggestions for Authors
The authors should be congratulated for their effort. I have the following comments:
-
Could you clarify why patients who died immediately after surgery were excluded from the study? This is a crucial postoperative outcome and should be considered as an important index for analysis.
-
Figure 1a does not seem to contribute meaningfully to the manuscript and could be omitted as it adds little to the overall content.
-
Figures 14 and 15 should be given more prominence in the manuscript, as they present key findings that should be central to your discussion.
-
Several grammatical and syntax issues were identified, including awkward phrasing and inconsistent terminology. Addressing these issues will enhance the clarity and professionalism of the manuscript.
Author Response
Comment 1. Could you clarify why patients who died immediately after surgery were excluded from the study? This is a crucial postoperative outcome and should be considered as an important index for analysis.
R: Patients who died immediately after surgery were included in the statistical analysis but the immunohistochemistry was not performed because of the difficulties of obtaining consent from family members. We made the corrections in lines 91-95.
Comment 2. Figure 1a does not seem to contribute meaningfully to the manuscript and could be omitted as it adds little to the overall content.
R: Thank you for your observation. Indeed, the information from Figure 1 was already in Table 1. We removed Figure 1a and renumbered Figure 1b as Figure 1.
Comment 3. Figures 14 and 15 should be given more prominence in the manuscript, as they present key findings that should be central to your discussion.
R: Thank you for the suggestion. The data regarding Figure 14 was discussed in lines 327-328. Regarding Figure 15, lines 411-433 contain the discussions regarding the prognosis of small bowel malignant tumors related to age, grading, positive lymph nodes, and tumor uniqueness (which represents the originality of our study).
Comment 4. Several grammatical and syntax issues were identified, including awkward phrasing and inconsistent terminology. Addressing these issues will enhance the clarity and professionalism of the manuscript.
R: Thank you for your observation. We performed another extensive verification and we corrected grammatical and syntax errors!
Reviewer 2 Report
Comments and Suggestions for Authors
thank you for allowing me to review this monocentric series on primary tumors of the small intestine. the major limitation of this series is to have included different histological types that respond to a very different natural history. i suggest that the authors refocus their series solely on adenocarcinomas, which would allow them to better emphasize their scientific messages and thus avoid bias.
Author Response
Comment 1. Thank you for allowing me to review this monocentric series on primary tumors of the small intestine. the major limitation of this series is to have included different histological types that respond to a very different natural history. I suggest that the authors refocus their series solely on adenocarcinomas, which would allow them to better emphasize their scientific messages and thus avoid bias.
R: Thank you for your suggestion. The heterogeneous nature of the small bowel malignant tumors (which includes epithelial and mesenchymal tumors, and also lymphomas and metastases), can make the statistical evaluation of the prognosis difficult and prone to bias. However, the purpose of our study was not only the evaluation of factors associated with survival but also of the pathological type of primitive malignant small bowel tumors. Moreover, the rarity of small bowel tumors (from which adenocarcinoma represents only less than half) makes the evaluation of the prognosis more difficult (and statistically not significant) than for all malignant small bowel tumors. Because small bowel adenocarcinoma represents a very rare disease, only multicentric studies (or meta-analyses including multiple studies) can attain statistical significance. We tried to perform a statistical analysis of survival (shown in the modified Figure 15), with similar results as for all patients (and also with a lack of statistical significance, because of the small number of patients).
Reviewer 3 Report
Comments and Suggestions for Authors
The incidence of small bowel tumors remains low. For diagnosis, CT-scan, enteroscopy and capsule endoscopy are very useful; unfortunately the last method is not practicable in emergency. Surgery is the best choice plus chemotherapy for some malignant tumors. Laparoscopic approach is feasible in selected cases.
The article is well written with clear inclusion and exclusion criteria, the group of patients at the limit of statistical processing (rare pathology). The anatomopathological and immunohistochemical diagnosis (mandatory today) is well documented. The discussions are relevant. The current bibliography could also include Romanian works published in national magazines.
Author Response
The incidence of small bowel tumors remains low. For diagnosis, CT-scan, enteroscopy and capsule endoscopy are very useful; unfortunately the last method is not practicable in emergency. Surgery is the best choice plus chemotherapy for some malignant tumors. Laparoscopic approach is feasible in selected cases.
The article is well written with clear inclusion and exclusion criteria, the group of patients at the limit of statistical processing (rare pathology). The anatomopathological and immunohistochemical diagnosis (mandatory today) is well documented. The discussions are relevant. The current bibliography could also include Romanian works published in national magazines.
R: Thank you for your appreciation! As a result of your suggestion, we added currently available studies from Romania (discussions in lines 434-450, references 9,34-38).
Round 2
Reviewer 1 Report
Comments and Suggestions for Authors
Authors adequately addressed my comments.